# Aberrant corticosteroid metabolism in tumor cells enables GR takeover in enzalutamide resistant prostate cancer

Jianneng Li[1], Mohammad Alyamani[1,2], Ao Zhang[1], Kai-Hsiung Chang[1], Michael Berk[1], Zhenfei Li[1], Ziqi Zhu[1], Marianne Petro[1], Cristina Magi-Galluzzi[3], Mary-Ellen Taplin[4], Jorge A Garcia[5], Kevin Courtney[6], Eric A Klein[7], Nima Sharifi[1,2,5,7]*

[1]Department of Cancer Biology, Lerner Research Institute, Cleveland, United States; [2]Department of Chemistry, Cleveland State University, Cleveland, United States; [3]Pathology and Laboratory Medicine Institute, Cleveland, United States; [4]Lank Center of Genitourinary Oncology, Dana-Farber Cancer Institute, Harvard Medical School, Boston, United States; [5]Department of Hematology and Oncology, Taussig Cancer Institute, Cleveland, United States; [6]Division of Hematology and Oncology, Department of Internal Medicine, University of Texas Southwestern Medical Center, Dallas, United States; [7]Department of Urology, Glickman Urological and Kidney Institute, Cleveland, United States

**Abstract** Prostate cancer is driven by androgen stimulation of the androgen receptor (AR). The next-generation AR antagonist, enzalutamide, prolongs survival, but resistance and lethal disease eventually prevail. Emerging data suggest that the glucocorticoid receptor (GR) is upregulated in this context, stimulating expression of AR-target genes that permit continued growth despite AR blockade. However, countering this mechanism by administration of GR antagonists is problematic because GR is essential for life. We show that enzalutamide treatment in human models of prostate cancer and patient tissues is accompanied by a ubiquitin E3-ligase, AMFR, mediating loss of 11$\beta$-hydroxysteroid dehydrogenase-2 (11$\beta$-HSD2), which otherwise inactivates cortisol, sustaining tumor cortisol concentrations to stimulate GR and enzalutamide resistance. Remarkably, reinstatement of 11$\beta$-HSD2 expression, or AMFR loss, reverses enzalutamide resistance in mouse xenograft tumors. Together, these findings reveal a surprising metabolic mechanism of enzalutamide resistance that may be targeted with a strategy that circumvents a requirement for systemic GR ablation.

*For correspondence: sharifn@ccf.org

## Introduction

Metastatic prostate cancer usually responds initially to medical or surgical castration, then eventually progresses as castration-resistant prostate cancer (CRPC), which is stimulated by intratumoral synthesis of testosterone and/or 5$\alpha$-dihydrotestosterone (DHT) and activation of the androgen receptor (AR) (*Attard et al., 2016*; *Chang et al., 2013*; *Mostaghel et al., 2014*; *Scher and Sawyers, 2005*; *Hearn et al., 2016*). Enzalutamide is a potent next-generation AR antagonist and prolongs survival for patients with metastatic CRPC (*Beer et al., 2014*; *Scher et al., 2012*; *Tran et al., 2009*). Unfortunately, enzalutamide resistance almost always emerges, eventually leading to disease lethality.

Emerging data suggest that potent AR inhibition with enzalutamide leads to a massive up-regulation of GR expression, which then permits the re-expression of about 50% of AR-responsive genes, in turn promoting tumor progression (*Arora et al., 2013*; *Isikbay et al., 2014*). A challenge has

been reconciling these findings with the therapeutic effects of glucocorticoids in CRPC (*Kach et al., 2015*; *Montgomery et al., 2014*; *Sartor et al., 2014*). Furthermore, targeting such a mechanism with GR antagonists may be problematic because total and systemic GR blockade is incompatible with life (*Sharifi, 2014*). Treatment directed at a tumor-specific mechanism that regulates GR would therefore be desirable. We hypothesized that similar to metabolic mechanisms that elicit DHT synthesis, which in turn stimulate AR in CRPC (*Chang et al., 2013*; *Knudsen and Penning, 2010*; *Sharifi, 2013*), a role for GR in enzalutamide resistance would be accompanied by a tumor metabolic switch that provides sustained tissue cortisol concentrations that enable GR activation. Such a scenario and mechanism may furnish a tumor-specific pharmacologic target and thereby avoid adverse effects associated with systemic GR ablation.

GR stimulation by cortisol in peripheral tissues is physiologically tightly regulated by 11$\beta$-hydroxysteroid dehydrogenase-2 (11$\beta$-HSD2), which enzymatically converts cortisol to inactive cortisone in humans and corticosterone to 11-dehydrocorticosterone in mice (*Figure 1A*) (*Chapman et al., 2013*). For example, fetal and placental 11$\beta$-HSD2 expression shields against maternal cortisol, thereby restricting GR stimulation and blocking premature fetal maturation (*Chapman et al., 2013*). Here, we show that enzalutamide resistance is marked by sustained cortisol concentrations in the prostate tumor that are attributable to a profound loss of 11$\beta$-HSD2 and impaired conversion to cortisone, which together de-repress GR and stimulate glucocorticoid-dependent signaling. Mechanistically, 11$\beta$-HSD2 loss is mediated by the ubiquitin E3-ligase autocrine mobility factor receptor (AMFR). Finally, sustained 11$\beta$-HSD2 expression reverses the metabolic phenotype of enzalutamide resistance and reinstates the therapeutic response to enzalutamide in vivo.

## Results

### Enzalutamide treatment triggers sustained cortisol levels

We determined the effect of enzalutamide treatment on metabolic conversion of [$^3$H]-cortisol to inactive cortisone in the LAPC4 and VCaP human cell line models of CRPC. Long but not short (i.e. 24 hr) enzalutamide exposure sustains cortisol levels by retarding conversion to cortisone in cells and media (*Figure 1B* and *Figure 1—figure supplement 1A*). Similarly, freshly harvested CRPC xenograft tumors grown in orchiectomized mice and treated with enzalutamide (*Tran et al., 2009*) have an impaired capability of inactivating [$^3$H]-cortisol by conversion to cortisone, when compared to tumors from mice treated with orchiectomy alone (*Figure 1C*). Enzalutamide suppresses LAPC4 viability in vitro (*Figure 1D*) in association with suppression of AR-regulated genes (*Figure 1—figure supplement 1B*). Growth recovers after sustained enzalutamide treatment (*Figure 1—figure supplement 1C*).

### Enzalutamide promotes 11β-HSD2 loss

To determine the mechanism underlying the metabolic phenotype of impeded cortisol inactivation, expression of 11$\beta$-HSD2 and 11$\beta$-HSD1, which catalyzes the reverse reaction, was assessed. Loss of 11$\beta$-HSD2 protein in AR-expressing LAPC4, VCaP and MDA-PCa-2b prostate cancer cell lines (but not an AR-negative prostate cancer cell line [*Figure 2—figure supplement 1A*]) was observed with enzalutamide treatment, while no consistent effect on 11$\beta$-HSD1 was detectable (*Figure 2A–D* and *Figure 2—figure supplement 1B*). No suppression of *HSD11B2* mRNA, which encodes 11$\beta$-HSD2, was observed with enzalutamide treatment, suggesting that 11$\beta$-HSD2 protein loss is not attributable to transcriptional suppression (*Figure 2—figure supplement 1C*) and no direct 11$\beta$-HSD2 antagonism by enzalutamide was observed (*Figure 2—figure supplement 1D*). Importantly, 11$\beta$-HSD2 loss is not attributable to GR stimulation (*Figure 2—figure supplement 1E*). To address the clinical significance of these observations, we interrogated prostate tissues obtained or derived from patients and treated with enzalutamide (*Figure 2E*). Nine of 11 tissues treated with enzalutamide obtained from patients with prostate cancer (two metastatic CRPC, two local prostate tissues from patients treated neoadjuvantly with enzalutamide and seven fresh tissues treated with enzalutamide ex vivo) had a loss of 11$\beta$-HSD2 with treatment. All post-treatment biopsies were obtained from patients who were maintained on enzalutamide treatment. Consistent with previously published observations (*Arora et al., 2013*), GR up-regulation was observed in a subset of clinical tissues (*Figure 2—figure supplement 1F*), all of which had 11$\beta$-HSD2 loss in *Figure 2E*. These findings

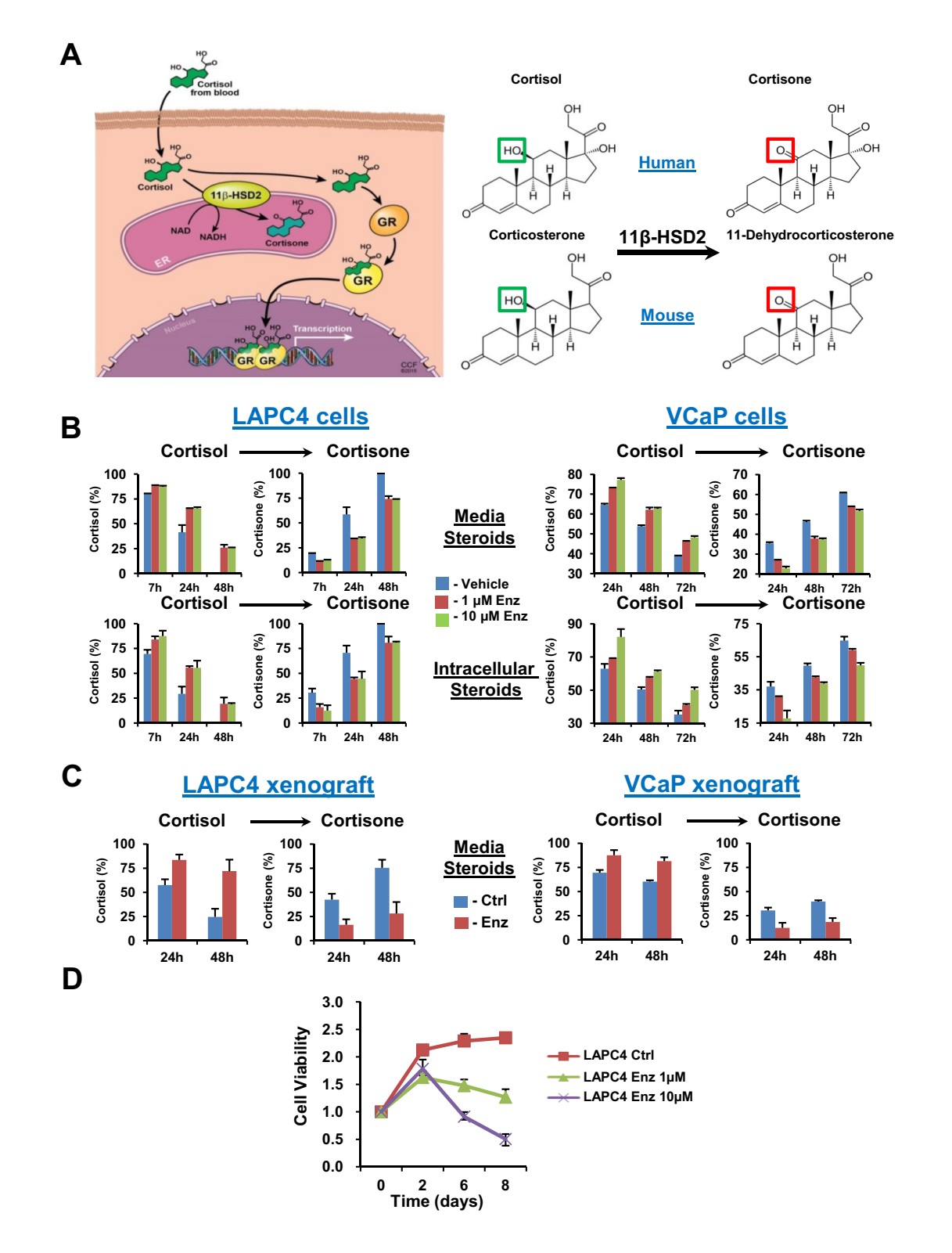

**Figure 1.** GR stimulation with enzalutamide resistance in prostate cancer is tightly regulated by glucocorticoid metabolism in target tissues. (A) Glucocorticoid metabolism in target tissues. Stimulation of GR by cortisol in humans is limited by 11β-HSD2, which oxidizes and converts cortisol to inactive cortisone. In mice, 11β-HSD2 converts active corticosterone to inactive 11-dehydrocorticosterone. (B) Enzalutamide (Enz) sustains cortisol levels by retarding inactivation in the LAPC4 and VCaP human prostate cancer cell lines. Cells were treated with the indicated concentrations of Enz or vehicle

*Figure 1 continued on next page*

*Figure 1 continued*

for 36 days (LAPC4) or 40 days (VCaP), and subsequently treated with [³H]-cortisol (100 nM) for the indicated times, followed by steroid extraction from media (above) and cells (below), steroid separation and quantitation with HPLC. The experiment was done in duplicate and repeated at least three times. (C) Cortisol inactivation is impaired in xenograft tumors treated with Enz. Fresh tumor tissues were harvested from LAPC4 or VCaP xenografts grown in orchiectomized mice and treated with Enz or chow alone (n = 5 tumors per treatment group). Tumors were treated with [³H]-cortisol (100 nM) for the indicated times and steroids were extracted from media and analyzed by HPLC. Error bars represent the SD. (D) Enz suppresses LAPC4 cell line proliferation. LAPC4 cells were treated with vehicle (Ctrl) or the indicated concentration of Enz for the designated number of days and cell viability was assessed using CellTiter-Glo. Cell viability was normalized to day 0, experiments were performed in triplicate and error bars represent the SD.

The following figure supplement is available for figure 1:

**Figure supplement 1.** Effects of Enz on LAPC4 cells.

demonstrate the potential clinical relevance of 11β-HSD2 protein loss for patients treated with enzalutamide.

## 11β-HSD2 reinstatement reverses the metabolic phenotype of enzalutamide treatment

In order to investigate the metabolic effects of 11β-HSD2 replacement on cortisol levels, we artificially expressed 11β-HSD2 in the context of enzalutamide exposure (*Figure 3A–B* and *Figure 3— figure supplement 1*). 11β-HSD2 reinstatement with both stable expression and transient transfection studies reverted the glucocorticoid metabolic phenotype of enzalutamide treated cells back to rapid cortisol inactivation that is characteristic of enzalutamide-naïve cells. Furthermore, the effects of 11β-HSD2 replacement on transcription of *PSA* (*Figure 3C*), which is regulated by AR and GR, *KLF9*, which is regulated by GR only and *PMEPA1*, which is regulated by AR only (*Arora et al., 2013*), suggest that the effects are indeed specific to glucocorticoid substrates of 11β-HSD2 (i.e., cortisol but not dexamethasone) and GR-responsive genes (i.e., *PSA* and *KLF9*).

## Reestablishing 11β-HSD2 expression restores sensitivity to enzalutamide therapy by depletion of active intratumoral glucocorticoids

We wished to determine if enzalutamide resistance is reversible with reinstatement of 11β-HSD2 expression. LAPC4 cells stably harboring a construct conferring forced 11β-HSD2 expression or vector alone (*Figure 4—figure supplement 1*) were injected subcutaneously and xenograft tumors were grown in surgically orchiectomized mice that were also implanted with sustained-release DHEA pellets to mimic the human adrenal androgen milieu in CRPC. When tumors reached 100 mm³, mice in each group were randomized to enzalutamide in chow or chow alone (*Figure 4A–B*). Tumors appeared to harbor significant resistance to treatment with enzalutamide, as evidenced by the growth of Vector xenografts through enzalutamide therapy. Forced 11β-HSD2 expression significantly reversed enzalutamide-resistant growth and prolonged progression-free survival. In contrast, 11β-HSD2 expression had no significant effect on untreated tumors, supporting a model in which the effect of 11β-HSD2 on tumor growth is specific to the context of resistance to potent AR antagonist therapy. Reinstatement of sensitivity to enzalutamide treatment also occurred in a second (VCaP) xenograft model of CRPC (*Figure 4C–D*). In contrast to humans, the mouse adrenal does not express 17α-hydroxylase/17,20 lyase and thus synthesizes corticosterone instead of cortisol as the dominant glucocorticoid (*Figure 1A*), which is similarly inactivated to 11-dehydrocorticosterone by 11β-HSD2 (*Miller and Auchus, 2011*). To validate if reversal of enzalutamide resistance with forced 11β-HSD2 expression is accompanied by the proposed intratumoral biochemical effect of depleting biologically active tumor glucocorticoids, we assessed corticosterone concentrations in enzalutamide-treated xenograft tissues by mass spectrometry (*Figure 4E*). 11β-HSD2 reinstatement depleted corticosterone concentrations by approximately two-thirds in enzalutamide-treated tumors (44.5 pmol/g in vector tumors vs. 15.1 pmol/g in 11β-HSD2 tumors; p=0.002), which otherwise harbor the capacity to metabolically sustain elevated concentrations of biologically active glucocorticoids. Tumor 11β-HSD2 expression also results in a significant decline in the percentage of tumor corticosterone (59% in vector tumors vs. 33% in 11β-HSD2 tumors; p<0.0001) when compared to

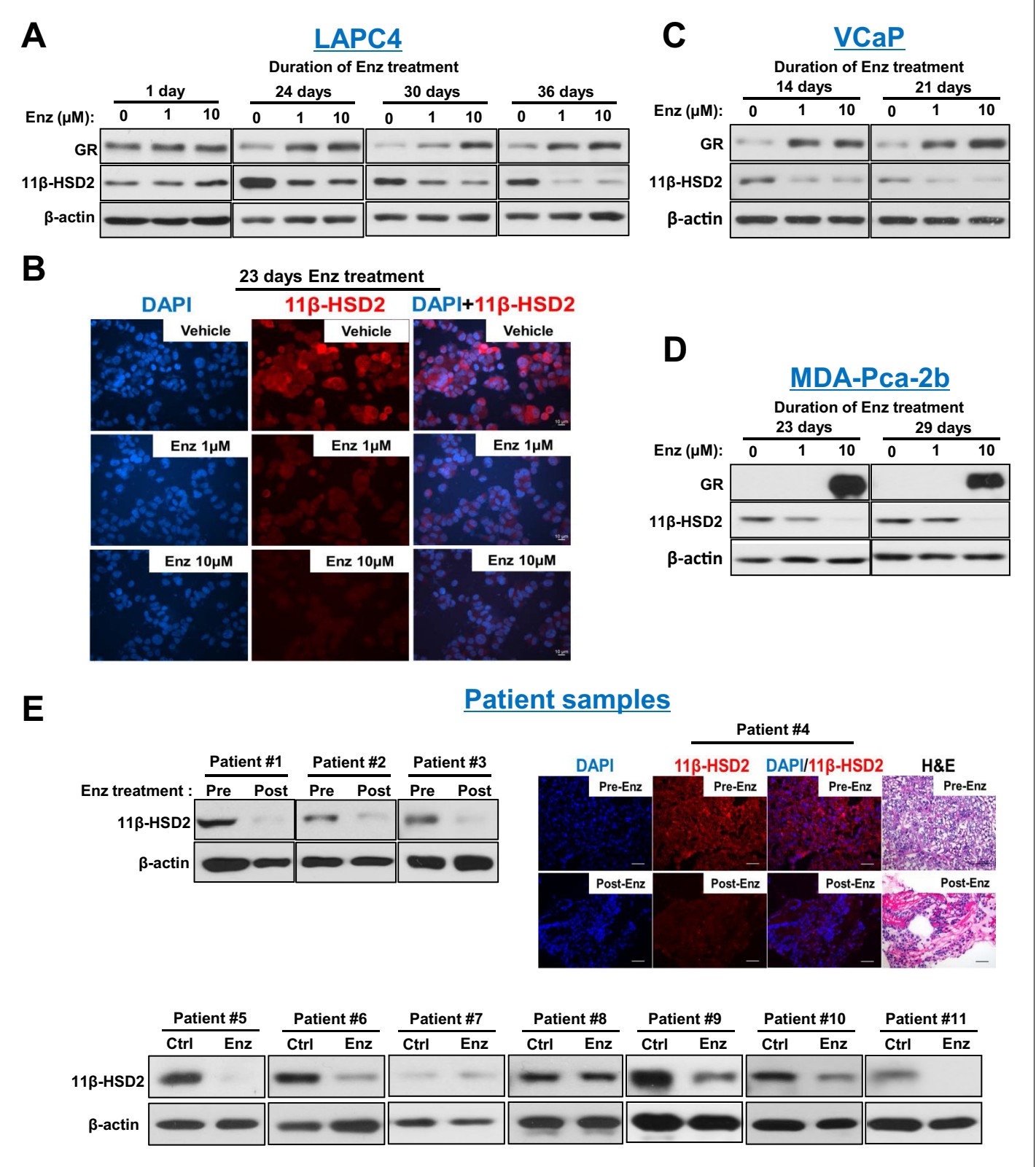

**Figure 2.** Enzalutamide promotes 11$\beta$-HSD2 protein loss in cell line models and tissues from patients with prostate cancer. (**A**) Enzalutamide (Enz) treatment results in the loss of 11$\beta$-HSD2 protein that occurs concurrently with an increase in GR protein in the LAPC4 model of CRPC as assessed by Western blot. (**B**) 11$\beta$-HSD2 protein expression in Enz treated LAPC4 cells as assessed by immunocytochemistry. (**C,D**) Loss of 11$\beta$-HSD2 and increase in GR protein similarly occur with Enz treatment in the VCaP and MDA-PCa-2b models. (**E**) Enz induces loss of 11$\beta$-HSD2 protein in tissue from patients

*Figure 2 continued on next page*

*Figure 2 continued*

with prostate cancer. Local prostate biopsies were obtained from Patients #1 and #2 with image guidance in a neoadjuvant study before (Pre) and after (Post) two months of treatment with Enz and medical castration. Patients #3 and #4 had biopsies of metastatic CRPC from lymph nodes before and after three months (Patient #3) and 11 months (Patient #4) of treatment with Enz. Fresh tissues from Patients #5-#11 were obtained from surgical prostatectomy specimens and incubated with vehicle or Enz (10 µM) for 7–8 days prior to protein extraction and Western blot.

The following figure supplement is available for figure 2:

**Figure supplement 1.** Response to Enz in prostate cancer cell lines and human tissues.

---

the sum total of corticosterone plus 11-dehydrocorticosterone (*Figure 4F*). In contrast to the effects of 11$\beta$-HSD2 expression on tumor glucocorticoids in enzalutamide treated mice, there is no significant change in serum in the absolute concentration (873 pmol/ml in vector mice vs. 867 pmol/ml in 11$\beta$-HSD2 mice; p=0.49; *Figure 4G*) or percentage of corticosterone (99.87% in vector mice vs. 99.85% in 11$\beta$-HSD2 mice; p<0.93; *Figure 4H*).

## AMFR is required for 11β-HSD2 ubiquitination, the metabolic phenotype of retarded glucocorticoid inactivation and enzalutamide resistance

We previously described the role of AMFR, a ubiquitin E3-ligase in the endoplasmic reticulum associated degradation pathway, in regulation of another steroidogenic enzyme, 3$\beta$-hydroxysteroid dehydrogenase-1 (*Chang et al., 2013*). As 11$\beta$-HSD2 is also located in the endoplasmic reticulum, we hypothesized that AMFR is required for enzalutamide-induced loss of 11$\beta$-HSD2 protein. A physical association between 11$\beta$-HSD2 and AMFR is supported by immunoprecipitation of AMFR, followed by immunoblot for 11$\beta$-HSD2, as well as immunoprecipitation of 11$\beta$-HSD2, followed by immunoblot for AMFR (*Figure 5A*). Expression of 11$\beta$-HSD2, Ubiquitin-His, and AMFR-Myc-DDK, followed by Ni-agarose pull-down and anti-11$\beta$-HSD2 immunoblot demonstrates the AMFR-dependence of 11$\beta$-HSD2 ubiquitination (*Figure 5B*). Enzalutamide treatment did not consistently increase AMFR expression (*Figure 5—figure supplement 1A–B*). However, Erlin-2, which enables the AMFR-associated endoplasmic reticulum-associated degradation pathway (ERAD), was more consistently up-regulated, including in 8 of 11 patient tissues (*Figure 5—figure supplement 1A–C*) (*Browman et al., 2006*; *Pearce et al., 2007*). The functional consequence of 11$\beta$-HSD2 / AMFR interaction and 11$\beta$-HSD2 ubiquitination is evidenced by genetically silencing AMFR with stable shRNA expression, which promotes an increase in 11$\beta$-HSD2 protein (*Figure 5C*) but not transcript (*Figure 5—figure supplement 1D*). Furthermore, the enzalutamide-induced metabolic phenotype that sustains cortisol concentrations by way of retarded inactivation is reversed with genetic ablation of AMFR (*Figure 5D*). Silencing both 11$\beta$-HSD2 and AMFR with enzalutamide treatment negates the effect of genetically silencing AMFR alone, suggesting that the effect of AMFR is mediated through 11$\beta$-HSD2 (*Figure 5—figure supplement 1E*). Finally, the functional relevance and requirement for AMFR expression on enzalutamide resistance is suggested by suppressed xenograft growth and prolonged progression-free survival in enzalutamide-treated xenografts with stable AMFR knockdown (*Figure 5E–F*). Tumors with AMFR knockdown were confirmed to have sustained 11$\beta$-HSD2 protein expression, thus impairing enzalutamide resistance (*Figure 5G*). Together, these findings suggest a model in which a physical association between AMFR and 11$\beta$-HSD2 enables enzalutamide to promote loss of 11$\beta$-HSD2, resulting in sustained cortisol concentrations that promote GR stimulation.

## Discussion

Our findings reveal a metabolic mechanism that is co-opted along with GR upregulation to stimulate enzalutamide resistance in prostate cancer. These findings indicate that systemic availability of GR agonists represents only one aspect of tumor GR stimulation in the setting of enzalutamide resistance. Local metabolic regulation of ligand availability by the tumor serves as a second critical aspect and can either oppose the effects of systemic glucocorticoids by spurring enzymatic inactivation, or instead allow unimpeded access to the tumor tissue, enabling sustained GR stimulation to promote tumor progression.

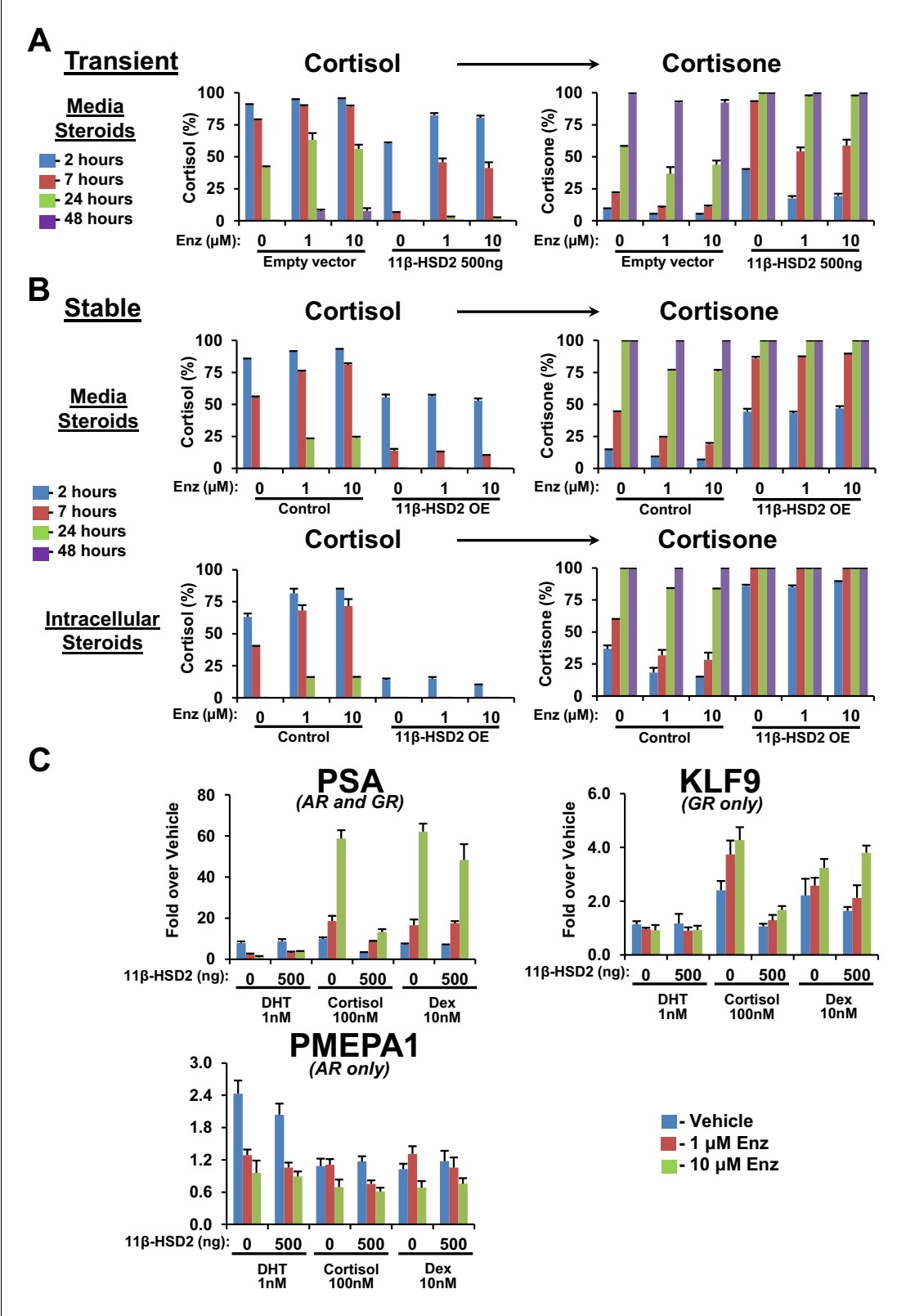

**Figure 3.** 11$\beta$-HSD2 expression reverses enzalutamide-sustained cortisol levels and GR-responsive gene expression. (**A**) Impeded conversion from cortisol to cortisone with Enz treatment is reversible with transient and (**B**) stable 11$\beta$-HSD2 expression. Cells expressing 11$\beta$-HSD2 or empty vector (control) were treated with the indicated concentration of Enz for 40 days, followed by treatment with [$^3$H]-cortisol and analysis of steroids in media by HPLC. (**C**) With Enz treatment, only cortisol-induced GR signaling is specifically reversible with forced stable 11$\beta$-HSD2 expression. LAPC4 cells were

*Figure 3 continued on next page*

*Figure 3 continued*

treated with Enz for 36 days, starved with phenol-red-free medium containing 5% Charcoal:Dextran-stripped FBS for 48 hr and transfected with a plasmid expressing 11$\beta$-HSD2 and treated with the indicated conditions for 24 hr. Only cortisol induction of *PSA* expression, which is GR- and metabolism-dependent, is reversible by 11$\beta$-HSD2. Expression of *KLF9*, which is regulated only by GR, is induced by cortisol and dexamethasone, but only cortisol induction is reversible by 11$\beta$-HSD2. Expression of *PMEPA1*, which is regulated only by AR, is induced with DHT only and not reversible by 11$\beta$-HSD2. Expression is normalized to vehicle-treated cells (not shown) and *RPLP0* expression. The experiment was performed four times. Error bars represent the SD of a representative experiment performed in triplicate.

The following figure supplement is available for figure 3:

**Figure supplement 1.** 11$\beta$-HSD2 overexpression (OE) in the long-term Enz-treated LAPC4 cells is comparable to endogenous expression in the human placental derived JEG-3 cell line.

Glucocorticoid administration has long been recognized to have a therapeutic effect in CRPC. Although the therapeutic effect of glucocorticoids is likely due in part to adrenal androgen suppression, the complete effects of glucocorticoids in prostate cancer have yet to be fully elucidated (*Attard et al., 2012*). Our observations move us closer to an understanding of the role of glucocorticoid pharmacology and physiology in prostate cancer. It has been recently recognized that GR stimulation may also contribute to prostate cancer progression. Our findings suggest yet another consideration that adds to the complexity of glucocorticoid signaling in prostate cancer, namely susceptibility of the administered glucocorticoid to metabolic inactivation, that is likely relevant to the increased GR expression that may occur alongside enzalutamide resistance and allows direct tumor-promoting effects of glucocorticoids in CRPC. For example, prednisolone is inactivated by 11$\beta$-HSD2 to prednisone but dexamethasone is generally thought to be impervious to inactivation by 11$\beta$-HSD2. This may be even more important prior to enzalutamide therapy and consequent suppression of 11$\beta$-HSD2 loss, because some data suggest that baseline GR expression prior to enzalutamide treatment, where 11$\beta$-HSD2 expression is intact, may be associated with enzalutamide resistance (*Arora et al., 2013*). Furthermore, early results from a neoadjuvant clinical trial of castration plus abiraterone suggest that tissue cortisol may be elevated specifically in prostate cancers that up-regulate GR (*Efstathiou et al., 2015*). Thus, it is possible that altered tumor glucocorticoid metabolism by way of the mechanism we have elucidated occurs not only with enzalutamide but also with abiraterone therapy.

Metabolic regulation of GR stimulation by the tumor might also be a tumor-specific therapeutic vulnerability. Our data raise the possibility that blocking 11$\beta$-HSD2 protein loss, for example by blocking AMFR, or reinstatement of 11$\beta$-HSD2 expression in the tumor may be an appropriate strategy to reverse enzalutamide resistance without affecting the systemic availability of glucocorticoids and resultant associated toxicities. Blocking AMFR may also increase 3$\beta$HSD1 protein, sustaining androgen synthesis (*Chang et al., 2013*). However, our in vivo studies with AMFR knockdown suggest that in the context of enzalutamide treatment, the net effect of AMFR ablation is therapeutic, probably because the AR ligand binding domain remains mainly occupied by enzalutamide and glucocorticoid signaling is a major driver of tumor progression.

Finally, our findings may have general relevance to steroid-dependent disease processes that use alternative steroid receptors. For example, in addition to the involvement of GR in prostate cancer, GR and AR have been implicated in a subtype-specific breast cancer progression (*Kach et al., 2015*; *Ni et al., 2011*). Our results suggest that a switch in steroid receptors that drives disease processes more broadly may be accompanied by perturbed local metabolic regulation of the availability of ligands that stimulate steroid receptor activation.

## Materials and methods

### Cell lines

LAPC4 was a generous gift from Dr. Charles Sawyers (Memorial Sloan Kettering Cancer, New York, NY), which was maintained in Iscove's modified Dulbecco's medium (IMDM) with 10% fetal bovine serum and incubated in a 5% $CO_2$ humidified incubator. VCaP was purchased from American Type

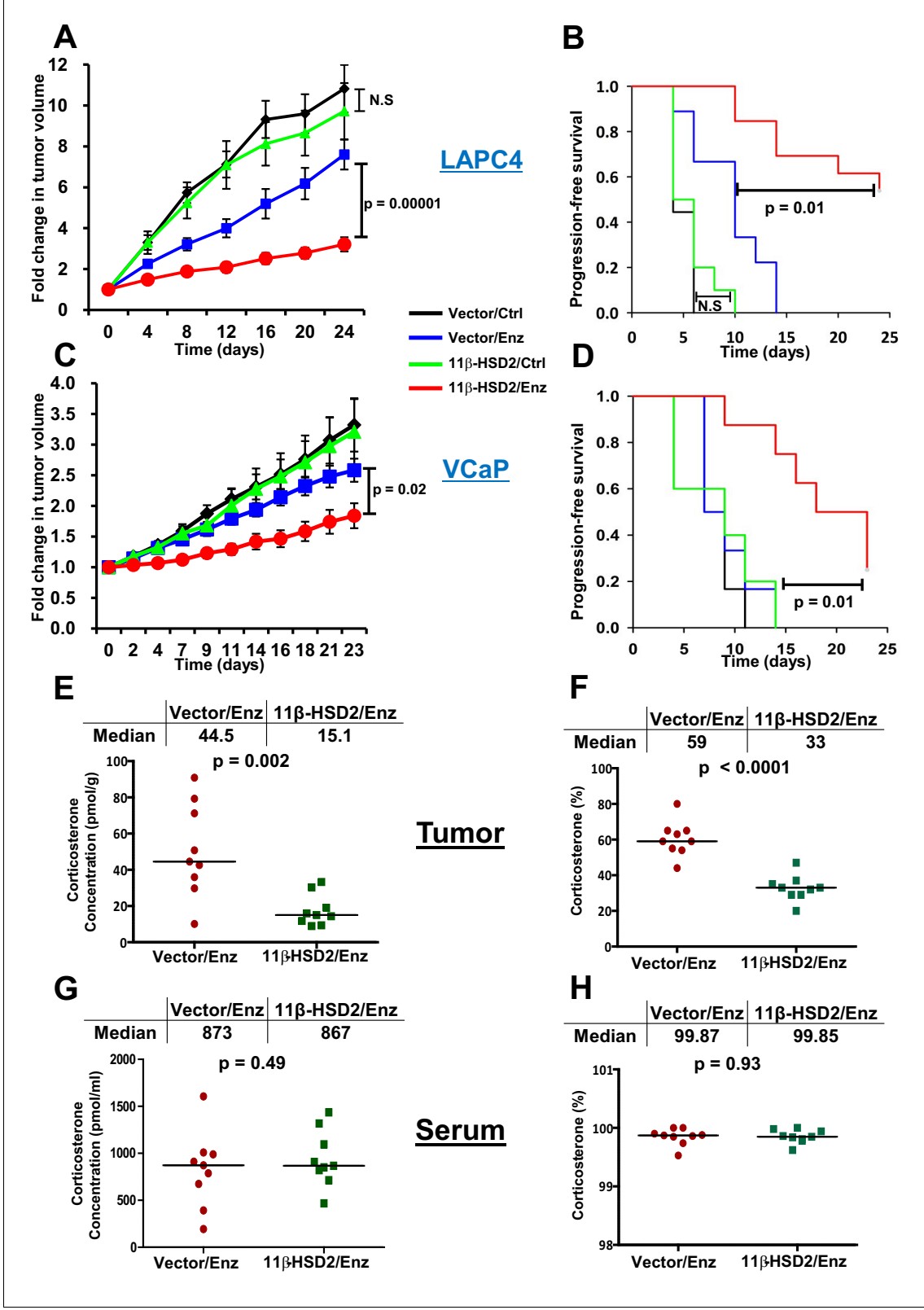

**Figure 4.** Reinstatement of 11β-HSD2 expression restores sensitivity to enzalutamide therapy by specifically suppressing tumor corticosterone. (A) Expression of 11β-HSD2 reverses enzalutamide (Enz) resistant LAPC4 CRPC xenograft tumor growth. (B) Progression-free survival is prolonged by 11β-HSD2 expression in Enz-treated LAPC4 xenografts. N.S. = not significant. (C) 11β-HSD2 expression reverses Enz resistance in the VCaP xenograft model of CRPC as assessed by decreased tumor volume and (D) prolongation of progression-free survival. For both xenograft studies, cells expressing 11β-
*Figure 4 continued on next page*

*Figure 4 continued*

HSD2 or vector (control) were grown in orchiectomized mice supplemented with DHEA and arbitrarily assigned to Enz or chow (Ctrl). For the comparisons in tumor volume, the significance of the difference between 11$\beta$-HSD2/Enz and Vector/Enz was calculated with an unpaired and two-tailed t-test on day 24 (LAPC4) or day 23 (VCaP). For the comparisons in progression-free survival, the significance of the difference between 11$\beta$-HSD2/Enz and Vector/Enz was calculated with a log-rank test. (E) The absolute concentration of corticosterone is reduced in xenograft tumors expressing 11$\beta$-HSD2. (F) The percentage of corticosterone relative to 11-dehydrocorticosterone is reduced in tumors expressing 11$\beta$-HSD2. (G) The absolute concentration of corticosterone and (H) percentage of corticosterone relative to 11-dehydrocorticosterone in serum are unaffected in mice harboring tumors with restored 11$\beta$-HSD2 expression. P values in E-H were calculated with an unpaired and two-tailed t-test.

The following figure supplement is available for figure 4:

**Figure supplement 1.** Forced 11$\beta$-HSD2 expression in Enz-treated LAPC4 xenografts is comparable to endogenous expression in the MDA-PCa-2b prostate cancer cell line and the human placental derived JEG-3 cell line.

Culture Collection (ATCC), which was cultured in Dulbecco's Modified Eagle Medium (DMEM) containing 10% fetal bovine serum and incubated in an 8% $CO_2$ humidified incubator. MDA-PCa-2b was purchased from ATCC, which was cultured in BRFF-HPC1 (Athena ES) containing 20% fetal bovine serum and incubated in a 5% $CO_2$ humidified incubator. FCIV1-11$\beta$-HSD2-FLAG (a gift from Moses Chao [Addgene plasmid # 24097]) (*Jeanneteau et al., 2008*) was used to generate the LAPC4 and VCaP stable cell line expressing human 11$\beta$-HSD2 by using a lentiviral system. The viral packaging and infection was performed as previously described (*Chang et al., 2011, 2013*). Briefly, 293T cells (ATCC) were cotransfected with 10 μg each of FCIV1-11$\beta$-HSD2-FLAG, pMD2.G, and psPAX2 vector for 48 hr to package the virus. Next, LAPC4 and VCaP cells were infected with the virus for 24 hr with the addition of polybrene (6 mg/ml), followed by selection with 2 ug/ml puromycin for ~2 weeks. The AMFR knockdown LAPC4 stable cell line was established by employing the pGFP-C-shLenti vectors contain AMFR shRNA sequences (5'-ACAAGACACCTCCTGTCCAACATGCAGAA-3' and 5'-GGAGCCGCTTCTCCAAGTCTGCTGATGAG-3') (OriGene), The viral packaging, infection as well as selection procedures were carried out as described above. Enzalutamide (Enz) was obtained courtesy of Medivation (San Francisco, CA). All Enz and vehicle treated cells were maintained in medium containing 10 nM DHEA. Cell lines are authenticated using short tandem repeat characterization by DDC Medical every six months and routinely (every 1–3 months) screened for mycoplasma contamination as described (*Li et al., 2015*).

## Cortisol metabolism

Cell line metabolism. Cells (~$10^6$ cells per well) were plated and maintained in 12 well plates coated with poly-DL-ornithine (Sigma-Aldrich) for overnight and then treated with [$^3$H]-cortisol (1,000,000 counts per minute (c.p.m.) per well; PerkinElmer) and non-radiolabeled cortisol (100 nM final concentration). After incubation for the indicated time points, both media and cells were collected. Briefly, 300 μl media was collected; the cells were scraped and centrifuged at 10,000 x g for 2 min twice to remove all the media, then the cell pellets were resuspended with 300 μl PBS. Collected media and cells were incubated with 300 units of $\beta$-glucuronidase (Helix pomatia; Sigma-Aldrich) at 65°C for at least 2 hr, extracted with 600 μl 1:1 ethyl acetate:isooctane, and concentrated under a nitrogen stream.

Xenograft metabolism. LAPC4 or VCaP cells (~$10^7$) were injected subcutaneously with Matrigel (Corning) into surgically orchiectomized NSG mice that were implanted with DHEA pellets (5 mg/pellet, 90-day sustained-release, Innovative Research of America). Fresh xenografts were harvested and ~40 mg xenograft tissues were minced, and cultured in IMEM with 10% FBS at 37°C with a mixture of [$^3$H]-cortisol and non-radiolabeled cortisol. Aliquots of media were collected at the indicated time points, steroids were extracted and concentrated as described above.

For HPLC analysis, the concentrated samples were dissolved in 50% methanol and injected on a Breeze 1525 system equipped with model 717 plus autoinjector (Waters Corp.). Steroid metabolites were separated by a Luna 150×4.6 mm, 3 μM C18 reverse-phase column (Phenomenex) using methanol/water gradients at 30°C. The column effluent was analyzed using a $\beta$-RAM model three in-line radioactivity detector (IN/US Systems, Inc.) using Liquiscint scintillation mixture (National Diagnostics). All metabolism studies were performed in duplicate and repeated in independent experiments.

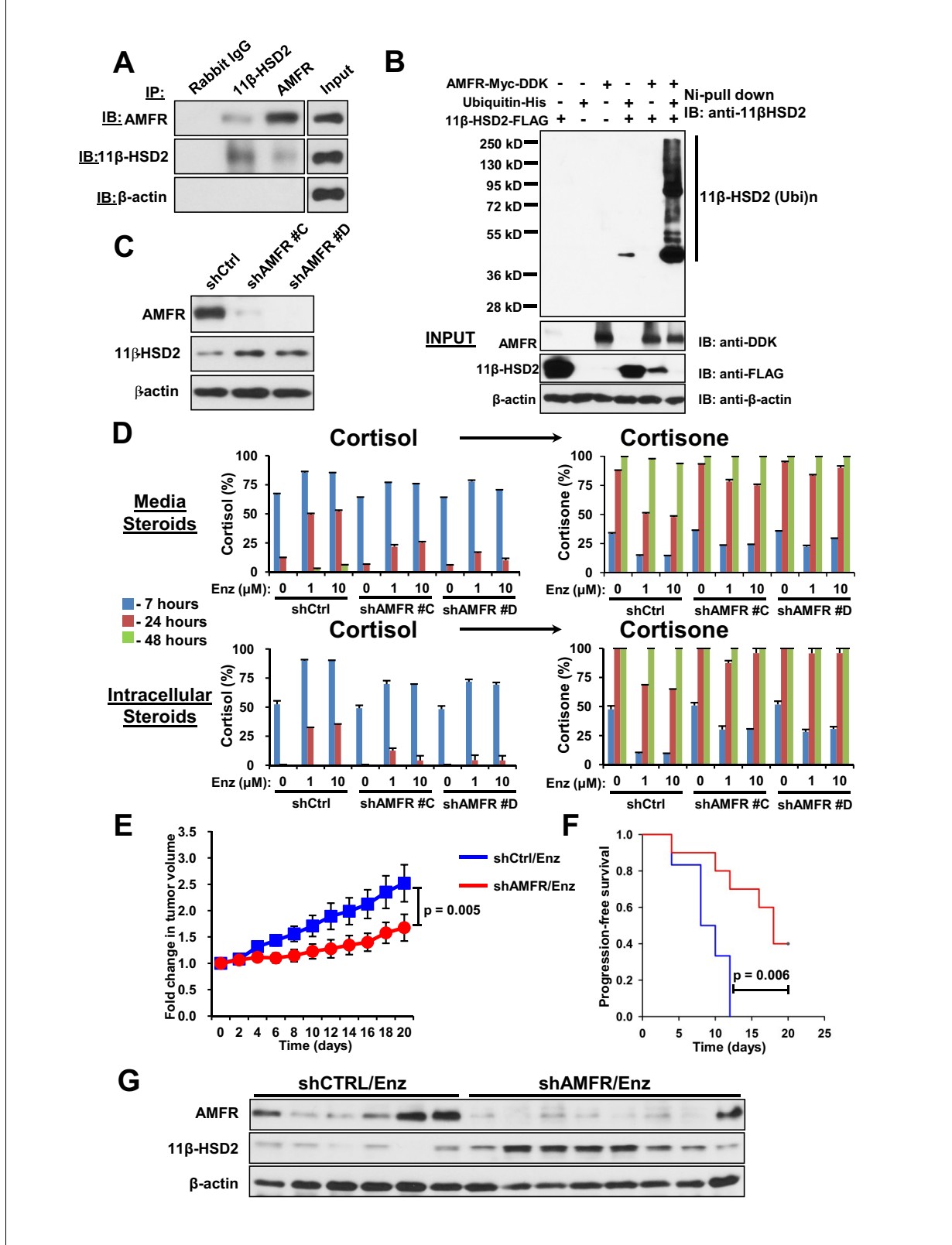

**Figure 5.** AMFR is required for 11β-HSD2 ubiquitination and the enzalutamide-induced metabolic phenotype that sustains local cortisol concentrations and enzalutamide-resistance. (**A**) 11β-HSD2 and AMFR co-immunoprecipitate. Immunoprecipitation (IP) and immunoblot (IB) from endogenously expressed proteins in whole cell protein lysate from LAPC4 cells were performed with the indicated antibodies. The experiment was performed twice. (**B**) AMFR promotes 11β-HSD2 ubiquitination. Proteins were expressed in 293 cells, proteins tagged with ubiquitin-His, were pulled-down with Ni-

*Figure 5 continued on next page*

*Figure 5 continued*

agarose beads, and immunoblot was performed with the indicated antibodies. The experiment was performed twice. (**C**) Silencing AMFR expression with two independent shRNAs increases 11$\beta$-HSD2 protein. LAPC4 cells stably expressed shRNAs against AMFR (shAMFR) or non-silencing control expression vector. The experiment was performed three times. (**D**) Blockade of Enzalutamide (Enz)-mediated 11$\beta$-HSD2 loss by silencing AMFR reverses the metabolic phenotype that confers sustained cortisol concentrations. Cells were treated with the indicated concentration of Enz and subsequently were treated with [³H]-cortisol (100 nM) for the indicated times, followed by steroid extraction from media and cells, and steroid analysis by HPLC. Error bars represent the SD of biological triplicates. The experiment was performed three times. Enz treatment in panel D was for 38–42 days. (**E**) AMFR is required for tumor growth through enzalutamide therapy. Xenografts from LAPC4 cells expressing shAMFR or non-silencing control vector (shCtrl) were grown in surgically orchiectomized mice supplemented with DHEA and treated with Enz when tumors reached 100 mm³. The significance of the difference between shCtrl and shAMFR groups was calculated with an unpaired and two-tailed t-test on day 20. (**F**) Progression-free survival is increased in tumors lacking AMFR. The significance of the difference between shCtrl and shAMFR groups was determined with a log-rank test. (**G**) Xenograft tumors with genetic ablation of AMFR retain 11$\beta$-HSD2 protein expression. Xenograft tissues were collected at the end of the xenograft study and immunoblot was performed with the indicated antibodies.

The following figure supplement is available for figure 5:

**Figure supplement 1.** AMFR and Erlin-2 regulation and cortisol metabolism with Enz treatment.

## Gene expression and immunoblot

LAPC4 cells were treated with Enz for 36 days and then seeded into 12 well plates coated with poly-DL-ornithine at 50% confluence. After incubation overnight, the cells were transfected with 11$\beta$-HSD2-FLAG plasmid for 48 hr by using the TransIT−2020 Transfection Reagent (Mirus) according to the protocol provided by the manufacturer, then maintained in phenol-red-free medium with 5% Charcoal:Dextran-stripped FBS for 48 hr before being treated with the indicated drugs. Total RNA was extracted with a GenElute Mammalian Total RNA miniprep kit (Sigma-Aldrich) and 1 μg RNA was reverse-transcribed to cDNA with the iScript cDNA Synthesis Kit (Bio-Rad). An ABI 7500 Real-Time PCR machine (Applied Biosystems) was used to perform the qPCR analysis, using iTaq Fast SYBR Green Supermix with ROX (Bio-Rad) in 96-well plates at a final reaction volume of 10 μl. The qPCR analysis was carried out in triplicate with the following primer sets: *PSA* (Forward: 5'-GCA TGGGATGGGGATGAAGTAAG-3'; Reverse: 5'-CATCAAATCTGAGGGTTGTCTGGA-3'), *FKBP5* (Forward: 5'-CCCCCTATTTTAATCGGAGTAC-3'; Reverse: 5'-TTTTGAAGAGCACAGAACACCT-3'), *TMPRSS2* (Forward: 5'-TGGTCCTGGATGATAAAAAAGTTT-3'; Reverse: 5'-GACATACGCCCCA-CAACAGA-3'), *GR* (Forward: 5'-CTAATGGCTATTCAAGCCCCAGCAT-3'; Reverse: 5'-GTGCTGTCC TTCCACTGCTCT-3'), *HSD11B2* (Forward: 5'-TGGATCGCGTTGTCCCG-3'; Reverse: 5'-GTTCAAC TCCAATACGGTGGC-3'), *HSD11B1* (Forward: 5'-GAGGTTCTCTCTGTGTGTCCT-3'; Reverse: 5'-G TAGTAGGCCATGAAGAGCCC-3'), *KLF9* (Forward: 5'-AACACGCCTCCGAAAAGAGG-3'; Reverse: 5'- CGTCTGAGCGGGAGAACTTT-3'), *PMEPA1* (Forward: 5'-GTGCAACTGCAAACGCTCTT-3'; Reverse: 5'- AGCTTGTAGTGGCTCAGCAG-3'), the housekeeping gene *large ribosomal protein P0* (*RPLP0*) (Forward: 5'-CGAGGGCACCTGGAAAAC-3'; Reverse: 5'-CACATTCCCCCGGATATGA-3'). Each mRNA transcript was quantitated by normalizing the sample values to *RPLP0* and to vehicle treated cells (for steroid treated cells). All gene expression studies were repeated in independent experiments.

For protein analysis, immunoblots were performed as described previously (*Li et al., 2013*). Briefly, total cellular protein was extracted with ice cold RIPA lysis buffer (Sigma-Aldrich) containing protease inhibitors (Roche). 30–50 μg protein was separated by 8% SDS-PAGE gel and then transferred to a nitrocellulose membrane (Millipore). After incubating with the anti-11$\beta$-HSD2 antibody (Santa Cruz; 1:3000), anti-11$\beta$-HSD1 antibody (Santa Cruz; 1:1000), GR antibody (BD Biosciences; 1:1000), anti-Erlin-2 antibody (Cell Signaling Technology: 1:1000) or anti-AMFR antibody (Protein-tech; 1:1000) overnight at 4°C, the appropriate secondary antibody was incubated for 1 hr at room temperature. The chemiluminescent detection system (Thermo Scientific) was used to detect the bands with peroxidase activity. An anti-$\beta$-actin antibody (Sigma-Aldrich; 1:5000) was used as a control for sample loading.

## Gene expression and knockdown

Gene expression. The day before transfection, cells were plated into 12 well plates coated with poly-DL-ornithine (~7 x 10$^5$ cells/well), then an Erlin-2 expressing plasmid, Erlin-2-Myc-DDK-tagged (OriGene) or 11$\beta$-HSD2 expressing plasmid FCIV1-11$\beta$-HSD2-FLAG was introduced into the cells with Lipofectamine 3000 Reagent (ThermoFisher, Waltham, MA). After 48 hr transfection, the cells were collected and used for the detection of 11$\beta$-HSD2 and Erlin-2 by immunoblot, or treated with Enz for 24 hr to determine the cortisol metabolism by HPLC as described above.

Gene knockdown. Cells were seeded into 12 well plates coated with poly-DL-ornithine at 60–80% confluence. After incubation overnight, the cells were transfected with siRNA following the Lipofect-amine RNAiMAX Reagent (ThermoFisher, Waltham, MA) protocol provided by the manufacturer for 48 hr. Cells were then used for qPCR and immunoblot analysis or cortisol metabolism analysis, as described previously. For 11$\beta$-HSD2 knockdown, the experiments were performed with Dhamarcon SMARTpool: ON-TARGETplus HSD11B2 siRNA, L-008983–00-0005 or ON-TARGET plus Non-target-ing Pool, #D-001810–10-05 with a final concentration of 25 nM siRNA. For Erlin-2 knockdown, the siRNA sequence: 5′-GCCTCTCCGGTACTAACAT-3′ (*Huber et al., 2013*) was used.

## Cell viability assay

LAPC4 cells or the long-term Enz treated LAPC4 cells were plated in triplicate in 96 well plates coated with poly-DL-ornithine and incubated overnight, then treated with Enz and assayed in tripli-cate at the time points indicated using CellTiter-Glo (Promega). Viability is normalized to day 0.

## Co-immunoprecipitation

The interaction between 11$\beta$-HSD2 and AMFR was analyzed using the Pierce Classic Magnetic IP/Co-IP Kit (Thermo Scientific) following the protocol provided by the manufacturer. Briefly, ~10$^7$ LAPC4 cells were lysed in 1 ml Pierce IP Lysis/Wash Buffer with protease inhibitors added fresh, on ice for 1 hr. The cell lysates were centrifuged at 12,000 x g for 15 min at 4℃. 1–2 mg of protein was pre-cleaned with 30 µl pre-cleaned Protein A/G PLUS-Agarose (Santa Cruz) and 1 µg rabbit IgG (Millipore) for 1 hr and then incubated with rabbit IgG (3 µg), 11$\beta$-HSD2 antibody (4 µg) or AMFR antibody (3 µg) overnight at 4℃. The antibody/antigen/bead complex was washed with ice-cold Pierce IP Lysis/Wash Buffer containing protease inhibitors adequately and denatured in 40 µl freshly prepared 1x Lane Marker Sample Buffer at room temperature for 30 min with mixing. 20 µl IP prod-ucts were used for the subsequent protein separation and detection of 11$\beta$-HSD2 or AMFR using their antibodies by immunoblot.

## Mouse xenograft studies

All mouse studies were performed under a protocol approved by the Institutional Animal Care and Use Committee (IACUC) of the Cleveland Clinic Lerner Research Institute. All NSG male mice (6–8 weeks old) were purchased from the Jackson Laboratory and the number of mice used in this study was based on previously published mouse xenograft studies by our lab that determined effects of steroid pathway inhibition/augmentation on xenograft growth (*Chang et al., 2011*, *2013*; *Li et al., 2015*). Mice were surgically orchiectomized and implanted with DHEA pellets to mimic human adre-nal DHEA production in patients with CRPC. one week later, mice were prepared for cell injections.

For the evaluation of the 11$\beta$-HSD2 role in reversing enzalutamide resistance, either 10$^7$ vector control or 10$^7$ 11$\beta$-HSD2 overexpressing LAPC4 or VCaP cells (100 µl in 50% matrigel and 50% growth media) were subcutaneously injected into mice. When tumors reached 100 or 150 mm$^3$ (length × width × height × 0.52), for LAPC4 and VCaP xenografts, respectively, the mice were arbi-trarily divided into two groups each for vector and 11$\beta$-HSD2 overexpressing cells: Enz diet 62.5 mg/kg and [as described in (*Tran et al., 2009*) or chow alone groups. Based on the daily chow con-sumption, approximately 0.3125 mg Enz was consumed per mouse per day. Enz in chow and chow alone were obtained from Medivation. Tumor volume was measured every other day, and progres-sion-free survival was assessed as time to 3-fold (LAPC4) or 1.5-fold (VCaP) increase in tumor volume (from 100 or 150 mm$^3$) from the time Enz or chow alone was initiated. The number of mice in the LAPC4 Vector/Ctrl, Vector/Enz, 11$\beta$-HSD2/Ctrl and 11$\beta$-HSD2/Enz groups were 9, 9, 10, and 11, respectively. The number of mice in the VCaP Vector/Ctrl, Vector/Enz, 11$\beta$-HSD2/Ctrl and 11$\beta$-HSD2/Enz groups were 6, 6, 5, and 8, respectively. Numbers of mice in each treatment group were

determined by those that survived surgical procedures and had reached a tumor volume to initiate treatment.

For evaluation of the role of AMFR in reversing enzalutamide resistance, either $10^7$ control or $10^7$ AMFR knockdown LAPC4 cells (100 µl in 50% matrigel and 50%growth media) were subcutaneously injected into mice. The remaining procedures were performed as described above. The number of mice in the LAPC4 shCtrl/Enz and shAMFR/Enz groups were 6, and 10, respectively. AMFR and 11$\beta$-HSD2 protein in the shCtrl and control or shAMFR LAPC4 xenografts were analyzed by immunoblot. Briefly, ~40–50 mg xenograft tissue was minced into pieces and then added into soft tissue homogenizing CK14 tubes (Betin Technologies) with 150 µl RIPA buffer containing protease inhibitors. Xenograft tissues were homogenized with a homogenizer (Minilys, Betin Technologies) six times (40 s each time) at the highest speed. Tubes were incubated on the ice for 5–10 min between each homogenization to cool lysates. The lysates were then centrifuged for 30 min at 15,000 x g and the supernatants were used for immunoblot analysis.

## Immunofluorescence staining

LAPC4 cells treated with Enz or vehicle for 23 days were seeded into chamber slides (BD Biosciences) coated with poly-DL-ornithine at 60% confluence. After overnight culture, cells were washed with PBS and fixed with ice cold methanol for 15 min and the methanol was washed with PBS. Before applying primary antibodies, nonspecific binding sites were blocked with blocking buffer (Protein Block Serum Free, Dako). Anti-11$\beta$-HSD2 antibody (Santa Cruz), diluted at 1:300 with Antibody Diluent (Dako), was applied for incubation overnight at 4°C. After being rinsed with PBS, the slides were probed with Alexa Fluor 594 conjugated secondary antibody (goat anti-rabbit, Thermo Scientific) for 45 min at room temperature. VECTASHIELD HardSet Mounting Medium (Vector Laboratories) was used to mount the slides and counterstain the nucleus with DAPI.

## Human tissue studies

All deidentified human tissues were obtained with informed consent using institutional review board (IRB)-approved protocols at each institution (Cleveland Clinic, UT Southwestern and Dana-Farber Cancer Institute). Pre- and post-Enz lymph node tissues were obtained from CT-guided biopsy of metastatic CRPC from Cleveland Clinic (Patient #3) and Dana-Farber Cancer Institute (Patient #4). Pathologic identification of tumor was done by an expert prostate cancer pathologist. Staining of tissues from Patient #4 was done with frozen section slides that were air dried at room temperature for 5 min, followed by rehydration with PBS. Immunofluorescence staining was performed as described above. H and E staining was completed by the imagine core of Biomedical Engineering Department in Lerner Research Institute of Cleveland Clinic.

Paired pre-Enz treatment and post-Enz treatment tissues from Patient #1 and Patient #2 were obtained from patients with localized prostate cancer treated with Enz plus ADT for two months prior to the second biopsy in a clinical trial (NCT02064582) at the University of Texas Southwestern Medical Center. Biopsies were obtained using image-guidance with a Koelis Urostation. The biopsy cores were minced into pieces and then added into Soft tissue homogenizing CK14 tubes (Betin Technologies) with 100 µl 6M Urea buffer containing protease inhibitors (Sigma Aldrich). Tissue homogenization and immunoblot analysis were performed as described previously.

Seven fresh prostate tissue cores (60–80 mg) from Patients #5-#11 were obtained from the peripheral zone of radical prostatectomy specimens at Cleveland Clinic, confirmed to have tumor in or in close proximity to cores by an expert prostate cancer pathologist, minced and aliquoted to two parts. One was treated with 10 nM DHEA plus vehicle, and the other was treated with 10 nM DHEA plus 10 µM Enz. Both tissues were maintained in 3 ml DMEM containing 10% fetal bovine serum and incubated in a 5% $CO_2$ humidified incubator. After four days of culture, two more ml medium with DHEA plus either vehicle or Enz was added into each part. The tissues were collected after 7–8 days treatment. The same procedures were performed as described above for protein extraction and immunoblot.

## Mass spectrometry

Xenograft analysis. At least 24 mg tumor tissue (n = 18) was homogenized with 1 ml LC-MS grade water (Fisher) by using homogenizer. The mixture was then centrifuged. 800 µl of the supernatant

was transferred to a glass tube, followed by the addition of 80 μl of 10 ng/ml internal standard (corticosterone-d8) (Steraloids). The steroids and the internal standard were extracted with methyl tert butyl ether (Across) evaporated to dryness under $N_2$ then reconstituted with 500 μl of 50% methanol.

Mouse serum analysis. At the endpoint of the xenograft study, mouse serum was collected. 20 μl of mouse serum and internal standard (corticosterone-d8) were precipitated with 200 μl methanol. After centrifugation, the supernatant was transferred to HPLC vials prior to mass spectrometry analysis.

The LC-MS/MS system contains an ultra-pressure liquid chromatography system (Shimadzu Corporation, Japan) which is consisted of two LC-30AD pumps, a DGU-20A5R degasser, a CTO-30A column oven, SIL-30AC autosampler, and a system controller CBM-20A and coupled with a Qtrap 5500 mass spectrometer (AB Sciex). Data acquisition and processing were performed using Analyst software (version 1.6.2) from ABSciex.

Steroids were ionized using electrospray ionization in positive mode. Quantification of analytes was performed using multiple reaction monitoring. The mass transitions for corticosterone, 11-dehydrocorticosterone, and internal standard are 347.3/121.0, 345.3/121.0, and 355.3/125.0, respectively. Separation of steroids was achieved using a Zorbax Eclipse plus C18 column (Agilent) using a mobile phase consisting of (A) 0.2% formic acid in water and (B) 0.2% formic acid in (methanol:acetonitrile, 60:40) with a gradient program at a flow rate of 0.2 ml/min. Sample injection volume was 10 μl.

## Ubiquitination assay

Experiments were conducted as previously described (*Chang et al., 2013*), with minor modifications. Briefly, HEK293T were transfected with the following plasmids: FCIV1-11$\beta$-HSD2-FLAG, pcDNA3-6xHis-ubiquitin and pLenti-AMFR-Myc-DDK (OriGene) for 36 hr. Transfected cells were collected by scraping with ice-cold PBS. Cell pellets were suspended in 200 μl PBS. For input analysis, 20 μl of cell suspension was pelleted and lysed with RIPA lysis buffer, followed by immunoblot analyses with anti-DDK (OriGene, 1:1000), anti-FLAG (Sigma-Aldrich, 1:1000) and anti-$\beta$-actin antibodies. The remaining cells were lysed with 4 ml lysis buffer (6 M guanidine-HCl, 0.1M $Na_2HPO_4$/$NaH_2PO_4$, 0.01MTris/HCl, pH 8.0, 10 mM imidazole, and 10 mM $\beta$-mercaptoethanol) and sonicated to reduce the viscosity. Protein complexes were pulled down by incubation with 30 μl Ni NTA magnetic agarose beads (QIAGEN) at room temperature for 2 hr and then successively washed with the buffer series: (1) 6 M guanidine-HCl, 0.1M $Na_2HPO_4$/$NaH_2PO_4$, 0.01M Tris/HCl, pH 8.0, 10 mM imidazole, and 10 mM $\beta$-mercaptoethanol; (2) 8 M Urea, 0.1 M $Na_2HPO_4$/$NaH_2PO_4$, 0.01 M Tris/HCl, pH 8.0, 20 mM imidazole, 10 mM $\beta$-mercaptoethanol plus 0.2% Triton X-100; (3) 8 M urea, 0.1 M $Na_2HPO_4$/$NaH_2PO_4$, 0.01 M Tris/HCl, pH 6.3, 10 mM $\beta$-mercaptoethanol (buffer A), 40 mM imidazole plus 0.4% Triton X-100, twice; (4) buffer A with 20 mM imidazole plus 0.2% Triton X-100; (5) buffer A with 10 mM imidazole plus 0.1% Triton X-100. After the washes, the protein complexes were eluted with 30 μl 2X SDS sample buffer containing 400 mM imidazole and 20 μl elution was then used for immunoblot analysis with anti-11$\beta$-HSD2 antibody (Santa Cruz; 1:3000).

## Acknowledgements

We thank David Schumick for help with the illustration in *Figure 1*. We thank Mike Brown for helpful comments. This work has been supported in part by funding to NS from a Howard Hughes Medical Institute Physician-Scientist Early Career Award, a Prostate Cancer Foundation Challenge Award, an American Cancer Society Research Scholar Award (12–038-01-CCE), grants from the National Cancer Institute (R01CA168899, R01CA172382, and R01CA190289).

## Additional information

### Funding

| Funder | Grant reference number | Author |
| --- | --- | --- |
| U.S. Department of Defense | W81XWH-16-1-0270 | Jianneng Li |

| | | |
|---|---|---|
| Prostate Cancer Foundation | | Zhenfei Li<br>Nima Sharifi |
| National Cancer Institute | R01CA168899 | Nima Sharifi |
| American Cancer Society | 12–038-01-CCE | Nima Sharifi |
| National Cancer Institute | R01CA172382 | Nima Sharifi |
| National Cancer Institute | R01CA190289 | Nima Sharifi |
| Howard Hughes Medical Institute | | Nima Sharifi |

The funders had no role in study design, data collection and interpretation, or the decision to submit the work for publication.

## Author contributions

JL, MA, AZ, NS, Conception and design, Acquisition of data, Analysis and interpretation of data, Drafting or revising the article; K-HC, ZL, ZZ, MP, Acquisition of data, Drafting or revising the article; MB, CM-G, Acquisition of data, Analysis and interpretation of data, Drafting or revising the article; M-ET, JAG, KC, EAK, Conception and design, Acquisition of data, Drafting or revising the article

## Author ORCIDs

Marianne Petro, http://orcid.org/0000-0003-3509-2999
Nima Sharifi, http://orcid.org/0000-0003-1281-3474

## Ethics

Human subjects: All deidentified human tissues were obtained with informed consent using institutional review board (IRB)-approved protocols at each institution (Cleveland Clinic, UT Southwestern and Dana-Farber Cancer Institute).
Animal experimentation: All animal work was done under Cleveland Clinic IACUC protocol number 2015-1549.

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
