## [Decision Letter]

Thank you for submitting your article "Aberrant tumor metabolism enables GR takeover in enzalutamide resistant prostate cancer" for consideration by *eLife*. Your article has been favorably evaluated by Charles Sawyers as the Senior Editor and three reviewers, including Ralph DeBerardinis (Reviewer #1) - who is a member of our Board of Reviewing Editors - and Charles Ryan (Reviewer #3).

I am pleased to say that the reviewers were positive about the paper. At this stage, the three reviewers have discussed the reviews with one another and the Reviewing Editor has drafted this decision to help you prepare a revised submission.

Summary of work:

Li et al. studied the role of glucocorticoid receptor (GR) expression in enzalutamide resistance in castration-resistant prostate cancer. This is an important question because up-regulation of GR and subsequent cortisol-mediated expression of a large number of androgen receptor responsive genes is believed to contribute to enzalutamide resistance. A key question is how to block cortisol's action locally, in the tumor, because systemic blockade of GR function is likely to be intolerably toxic. Here the authors show that enzalutamide causes a delay of cortisol conversion to cortisone in CPRC cell lines and xenografts. This is associated with increased GR expression and decreased expression of 11β-HSD2, a cortisol-metabolizing enzyme, in cells, primary tumor tissue and tumors treated ex vivo with enzalutamide. Enforced 11β-HSD2 expression is sufficient to decrease residual cortisol levels and decrease expression of cortisol-responsive genes in cell lines. It also reduces cortisosterone levels in xenografts and enhances the therapeutic effect of enzalutamide. Mechanistically, 11β-HSD2 physically interacts with AMFR, an ER-associated E3-ubiquitin ligase, and eliminating AMFR expression enhances 11β-HSD2 expression, increases conversion of cortisol to cortisone, and enhances the effect of enzalutamide on tumor growth. Overall, the authors provide evidence for a new concept in counteracting enzalutamide resistance by reducing local GR signaling in CRPC tumors.

Essential Revisions:

1) Enzalutamide's effects on cortisol metabolism are largely assessed in LAPC4 cells. Do these effects appear in some of other systems featured in the paper (other cell lines and/or prostate cancer samples treated ex vivo)?

2) The mechanism of reduced 11β-HSD2 expression in enzalutamide-treated cells should be better defined. The data showing that Erlin-2 increases in enzalutamide-resistant cells is intriguing, but should be functionally linked to resistance. The authors should test whether Erlin-2 over-expression is sufficient to decrease cortisol metabolism and/or whether silencing Erlin-2 after long-term enzalutamide treatment potentiates conversion of cortisol to cortisone.

3) If the mechanism of Dex and Pred's clinical effects in CRPC is solely through the suppression of adrenal steroid production, how can the investigators claim that altered Dex and Pred metabolism within a tumor cell mediates resistance? They might be better served by suggesting that Dex and Pred exert their clinical benefits in as-yet-unspecified mechanism, and that the new observations in the paper move us closer to understanding this. It may well be true that corticosteroids do not exert all of their clinical effects through suppression of adrenal androgens. The authors cite the Attard data but the interesting thing about these data is that patients with progression on abiraterone (with maximal adrenal androgen suppression already achieved) may still respond to dexamethasone. Thus, steroids may exert both clinical benefits in some settings but may also mediate resistance.

4) The in vitro experiments are elegant and sufficiently support the hypothesis that alterations in 11β-HSD2 are mechanistically responsible for sustained glucocorticoid levels in the tumor cells. Do the authors think that this effect is specific to enzalutamide, a class effect of AR targeted drugs, or possibly a broader mechanism of CRPC development?

Title: For improved clarity, we suggest changing the title to "Aberrant corticosteroid metabolism in tumor cells enables GR takeover in enzalutamide resistant prostate cancer."

---

## [Author Response]

*[…] Essential Revisions:*

*1) Enzalutamide's effects on cortisol metabolism are largely assessed in LAPC4 cells. Do these effects appear in some of other systems featured in the paper (other cell lines and/or prostate cancer samples treated ex vivo)?*

These effects also occur in the VCaP model. We have now included additional in vitroand xenograft data with the VCaP model in Figure 1.

*2) The mechanism of reduced 11β-HSD2 expression in enzalutamide-treated cells should be better defined. The data showing that Erlin-2 increases in enzalutamide-resistant cells is intriguing, but should be functionally linked to resistance. The authors should test whether Erlin-2 over-expression is sufficient to decrease cortisol metabolism and/or whether silencing Erlin-2 after long-term enzalutamide treatment potentiates conversion of cortisol to cortisone.*

We have performed additional experiments knocking down Erlin-2 in enzalutamide-treated LAPC4 cells. These data show that Erlin-2 knockdown increases conversion from cortisol to cortisone, effectively reversing the metabolic phenotype conferred by enzalutamide treatment (Figure 5—figure supplement 1).

*3) If the mechanism of Dex and Pred's clinical effects in CRPC is solely through the suppression of adrenal steroid production, how can the investigators claim that altered Dex and Pred metabolism within a tumor cell mediates resistance? They might be better served by suggesting that Dex and Pred exert their clinical benefits in as-yet-unspecified mechanism, and that the new observations in the paper move us closer to understanding this. It may well be true that corticosteroids do not exert all of their clinical effects through suppression of adrenal androgens. The authors cite the Attard data but the interesting thing about these data is that patients with progression on abiraterone (with maximal adrenal androgen suppression already achieved) may still respond to dexamethasone. Thus, steroids may exert both clinical benefits in some settings but may also mediate resistance.*

We fully agree with these comments. We have changed this section of the Discussion to read, “Although the therapeutic effect of glucocorticoids is likely due in part to adrenal androgen suppression, the complete effects of glucocorticoids in prostate cancer have yet to be fully elucidated. Our observations move us closer to an understanding of the role of glucocorticoid pharmacology and physiology in prostate cancer.”

*4) The in vitro experiments are elegant and sufficiently support the hypothesis that alterations in 11β-HSD2 are mechanistically responsible for sustained glucocorticoid levels in the tumor cells. Do the authors think that this effect is specific to enzalutamide, a class effect of AR targeted drugs, or possibly a broader mechanism of CRPC development?*

Sustained prostate tissue glucocorticoid concentrations may also occur with abiraterone therapy, as is suggested by early results of a neoadjuvant clinical trial of medical castration plus abiraterone acetate. We think it is likely those early clinical results may be attributable to the same mechanism we describe in this manuscript, although this would have to be investigated directly. We have included a discussion of this clinical trial in the second paragraph of the manuscript Discussion.

Title: For improved clarity, we suggest changing the title to "Aberrant corticosteroid metabolism in tumor cells enables GR takeover in enzalutamide resistant prostate cancer."

Thank you for this suggestion. We have made this change to the manuscript title.